# Next-Generation CEA-CAR-NK-92 Cells against Solid Tumors: Overcoming Tumor Microenvironment Challenges in Colorectal Cancer

**DOI:** 10.3390/cancers16020388

**Published:** 2024-01-16

**Authors:** Alexander Sebastian Franzén, Abdelhadi Boulifa, Clarissa Radecke, Sebastian Stintzing, Martin J. Raftery, Gabriele Pecher

**Affiliations:** 1Berlin Institute of Health at Charité, Universitätsmedizin Berlin, Charitéplatz 1, 10117 Berlin, Germany; 2Competence Center of Immuno-Oncology and Translational Cell Therapy (KITZ), Department of Hematology, Oncology and Tumor Immunology, CCM, Charité–Universitätsmedizin Berlin, Corporate Member of Freie Universität Berlin and Humboldt-Universität zu Berlin, Charitéplatz 1, 10117 Berlin, Germany

**Keywords:** immunotherapy, CAR-NK, CEA, PD1-checkpoint inhibition (CPI), CCR4, colorectal carcinoma (CRC), NK-92-cell line, tumor microenvironment, trogocytosis, 3D tumor models

## Abstract

**Simple Summary:**

Colon cancer is a solid tumor that is a prominent contributor to global mortality. Immune cells genetically engineered with a chimeric antigen receptor (CAR) that can recognize cancer-specific targets is a new innovative therapy approach that has had success in treating blood cancers but is still in development for treating solid tumors such as colon cancer. Part of the reason for the added difficulty in targeting solid tumors is the tumor microenvironment that acts as a protective barrier around a solid tumor. In this research paper, we have developed a new cellular approach for the targeted treatment of colon cancer that is designed to overcome the tumor microenvironment. We tested this new CAR cell therapy against multiple solid colon cancer models, and confirmed its efficacy and functionality in finding and eliminating solid tumors.

**Abstract:**

Colorectal carcinoma (CRC) presents a formidable medical challenge, demanding innovative therapeutic strategies. Chimeric antigen receptor (CAR) natural killer (NK) cell therapy has emerged as a promising alternative to CAR T-cell therapy for cancer. A suitable tumor antigen target on CRC is carcinoembryonic antigen (CEA), given its widespread expression and role in tumorigenesis and metastasis. CEA is known to be prolifically shed from tumor cells in a soluble form, thus hindering CAR recognition of tumors and migration through the TME. We have developed a next-generation CAR construct exclusively targeting cell-associated CEA, incorporating a PD1-checkpoint inhibitor and a CCR4 chemokine receptor to enhance homing and infiltration of the CAR-NK-92 cell line through the TME, and which does not induce fratricidal killing of CAR-NK-92-cells. To evaluate this therapeutic approach, we harnessed intricate 3D multicellular tumor spheroid models (MCTS), which emulate key elements of the TME. Our results demonstrate the effective cytotoxicity of CEA-CAR-NK-92 cells against CRC in colorectal cell lines and MCTS models. Importantly, minimal off-target activity against non-cancerous cell lines underscores the precision of this therapy. Furthermore, the integration of the CCR4 migration receptor augments homing by recognizing target ligands, CCL17 and CCL22. Notably, our CAR design results in no significant trogocytosis-induced fratricide. In summary, the proposed CEA-targeting CAR-NK cell therapy could offer a promising solution for CRC treatment, combining precision and efficacy in a tailored approach.

## 1. Introduction

In recent years, immunotherapy has emerged as a promising approach to combat various forms of cancer [1]. Among the different immunotherapeutic strategies, natural killer (NK) cell-based therapies have gained significant attention. NK cells possess unique characteristics that make them highly suitable for cancer treatment. They can recognize and eliminate malignant cells without prior sensitization, providing an innate immune response against cancer [2,3]. Additionally, NK cells offer the advantage of being an allogenic off-the-shelf product, eliminating the need for patient-specific preparations [1,2,3].

Compared to T-cells, NK cells do not produce graft-versus-host disease (GvHD), making them safer for use in allogenic therapies [4,5,6]. Moreover, their innate Fc-gamma receptor enables effective combination with targeted antibodies, harnessing the power of antibody-dependent cellular cytotoxicity (ADCC) to elicit a potent anti-tumor response [7,8].

One promising approach in NK cell-based immunotherapy is the engineering of chimeric antigen receptors (CARs) in NK cells. This strategy has shown immense potential in enhancing NK cell’s antitumor activity by enabling them to target specific cancer antigens [9,10]. Carcinoembryonic antigen-related cell adhesion molecule 5 (CEA) is one such target of interest. CEA is a glycoprotein overexpressed in various epithelial tumors, including pancreatic, breast, lung, and colon cancer. Its functional association with tumor differentiation, invasion, and metastasis makes it an attractive target for cancer treatment [11].

A critical consideration in NK cell therapy is the source of cells for the CAR product. Common sources include iPSC, cord blood, buffy coat, leukapheresis, or NK cell lines [7,9,12,13,14]. Each source presents different hurdles to overcome. Patient-derived NK cells face challenges related to donor variability, difficulty in genetic engineering and expansion of NK cells to therapeutic dosages without the use of feeder cells [15,16]. Therefore, donor-derived CAR NK products have a high batch-to-batch variability and are difficult to standardize. NK cell lines have been explored as an alternative source to patient-derived NK cells. They are easy to genetically engineer, can grow indefinitely, and exhibit high targeted cytotoxic potential when engineered with a CAR [17,18]. Thus, standardization of the CAR product can easily be achieved with NK cell lines. NK-92 cells are, however, devoid of the Fc-gamma receptor which could limit their synergistic utility when combined with monoclonal antibody-based immunotherapies [19]. In this study, we employed the NK-92 cell line as effector cells which have been shown to be safe in clinical trials when extra safety measures are taken such as irradiation before administration due to their malignant origin [17,18,20].

The development of effective cellular immunotherapy against solid tumors has encountered significant challenges due to the immunosuppressive nature of the tumor microenvironment (TME). Comprehensive understanding of the TME in solid tumors has been impeded by the absence of preclinical models that accurately capture tumor heterogeneity, replicate complex TME interactions, and facilitate high-throughput drug screening [21,22,23,24]. Patient-derived xenograft explant mouse models are regarded as the gold standard for preclinical testing [25,26]. However, their labor-intensive nature, ethical implications and incompatibility with high-throughput screening limit their utility. While traditional 2D cell culture has provided valuable insights, it fails to fully encompass the intricate in vivo interactions, particularly for cellular immunotherapies that are influenced by TME factors such as nutrient deprivation, metabolic byproduct accumulation, hypoxia, pH, and physical stromal barriers [23,27,28,29,30]. To address this limitation, 3D multicellular tumor spheroid (MCTS) models have been developed to better mimic tumor conditions, including cell-to-cell and cell-to-extracellular matrix interactions, pH and nutrient barriers, and allowing for high-throughput screening [24,29,31]. MCTS models offer a more physiologically relevant representation of the in vivo environment compared to standard 2D cell culture, as well as providing a more therapy-relevant platform for drug testing [32,33]. CRC MCTS, furthermore, show closer gene profiles to Xenograft models than 2D models, further strengthening the suitability of this model for drug testing [30]. 

Here, we used multiple in vitro solid tumor models with various layers of complexity to validate the effectiveness of a next-generation NK-cell CAR targeting CEA, utilizing a CCR4 receptor to increase homing and migration towards the tumor tissue. Our findings demonstrate the effectiveness of the CEA-CAR in eliminating multiple solid tumor models, while maintaining low activity against non-tumor tissue. The CEA-CAR did not cause any increase in CAR-mediated trogocytosis or fratricide, underlining the safety and efficacy of the CAR, and the CCR4 receptor enabled migration towards the tumor.

## 2. Materials and Methods

### 2.1. Cell Lines and Cell Culture

The colon carcinoma cell lines LS174T (DSMZ, Braunschweig, Germany), HT-29 (DSMZ), SW948 (DSMZ), SW1222 (ECACC, Salisbury, UK), and SW1417 (ATCC, Gaithersburg, MD, USA) were maintained in Dulbecco’s Modified Eagle Medium (DMEM; Gibco, Paisley, UK), supplemented with 10% inactivated fetal bovine serum (FBS; Gibco), 200 mM L-Glutamine, non-essential amino acids (NEAA; Gibco), and 1% penicillin/streptomycin (Pen/Strep; Gibco). The fibroblast cell line BJ (ATCC) was cultured in DMEM supplemented with 10% FBS (FBS; Gibco). The Jurkat reporter cell lines (supplied by Peter Steinberger, Medical University of Vienna, Austria) were maintained in Roswell Park Memorial Institute (RPMI) 1640 medium (Gibco) supplemented with 10% inactivated fetal bovine serum (FBS; Gibco), 200 mM L-Glutamine, and 1% penicillin/streptomycin (Pen/Strep; Gibco). Human embryonic kidney (HEK) 293T (DSMZ) cells were cultured in Dulbecco’s modified Eagle’s medium (DMEM; Gibco) supplemented with 10% FBS, 200 mM L-Glutamine, and 1% penicillin/streptomycin (Pen/Strep; Gibco). The YT-cell line (DSMZ, Germany) was cultured in RPMI-1640 supplemented with 10% FBS, 10 U/mL IL-2 (Immunotools, Friesoythe, Germany), 200 mM L-Glutamine, and 1% penicillin/streptomycin (Pen/Strep; Gibco). NK-92 cell line (DSMZ, Germany) was maintained in RPMI-1640 supplemented with 10% FBS, 200 U/mL IL-2 (Immunotools), 200 mM L-Glutamine, 1× NEAA (Gibco), 1× sodium pyruvate, and 1% penicillin/streptomycin (Pen/Strep; Gibco). PBMCs were isolated from buffy coats (DRK, German Red Cross; Dresden, Germany) using density gradient centrifugation and were further cultivated in RPMI, 10% FBS. Activation was followed by stimulating PBMCs with phytohaemagglutinin (PHA; Sigma-Aldrich, Darmstadt, Germany) and further maintaining them in medium with enriched with 50 units/mL IL2 (Immunotools). All cells were cultivated in a humidified cell culture incubator at 37 °C, 5% CO_2_, and regularly tested for mycoplasma (MycoAlert Lonza, Köln, Germany).

### 2.2. Multicellular Tumor Spheroid Culture (3D Cell Culture)

Three-dimensional (3D) cell culture was established using a liquid overlay technique by seeding the cells of interest into a pre-coated 96-well plate coated with anti-adherence solution (Stemcell Technologies, Köln, Germany). The plates were subsequently centrifuged at 200× *g* for 5 min and incubated in a humidified cell culture incubator at 37 °C and 5% CO_2_ for 4 days to promote multicellular tumor spheroid (MCTS) formation. In the case of co-culture MCTS, a 1:1 ratio of tumor cell to fibroblasts was used. For triple culture MCTS, a 1:1:0.5 ratio was used (LS174T: BJ: PBMC); additionally, the triple cultures were used for cytotoxicity experiments after 48 h. MCTS formation and cultivation was performed in DMEM supplemented with 10% FBS, 200 mM L-Glutamine, 1% NEAA, and 1% pen/strep. When MCTS were applied in cytotoxicity assays, there were no medium changes performed in the MCTS cultures to allow for tumor-conditioned medium formation.

### 2.3. Lentivirus Vector Production and Transduction 

Lentivirus production was achieved by transient transfection of HEK-293T cells using polyethylenimine (PEI). The harvested virus was titrated to determine the multiplicity of infection (MOI) and then flash frozen for storage at −80 °C. For transduction, the target cells were resuspended in cell culture medium containing the prepared virus and polybrene (5 µg/mL). After resuspension, the cell suspension was subjected to incubation on a spinning rotator at room temperature (RT) for 60 min, followed by a spin inoculation step at RT at 800× *g* for 120 min. The cells were then incubated overnight under standard cell culture conditions. The next day, the transduction media were replaced with the appropriate media for further cultivation.

### 2.4. Cytotoxicity Assays 

The CellTiter Glo^®^ 2.0 and CellTiter Glo^®^ 3D cell viability assay systems (Promega, Madison, WI, USA) were used to quantify cytotoxicity by measuring metabolic active cells through ATP presence through a luminescent luciferase reaction that is directly proportional to viable cells in the culture. Briefly, target cells and effector cells were co-cultured in the indicated E:T ratios, cell numbers, treatments, and time points. At the experimental end point, the viability assay reagent was added to the co culture in a 1:1 ratio (total culture volume–assay reagent) followed by shaking of the assay plate, incubation for 10 min (2D) or 25 min (3D) at room temperature, and subsequently measured using a Tristar 3 multimode plate reader (Berthold Technologies, Bad Wildbad, Germany). Target cell lysis was quantified using the formula: $$\mathrm{Cytotoxicity}\;(\%)=100-(\frac{Sample\;release-Effector\;cell\;release}{Maximum\;release})\times 100$$

### 2.5. Migration Assay

The migration potential of the transduced NK cell lines was determined utilizing cell culture inserts (ThinCert^®^; Greiner Bio-One, Frickenhausen, Germany) with an 8 µm pore size. A fixed number of 50,000 cells were loaded in the cell culture inserts at a total volume of 100 μL assay medium without chemokines. The inserts were placed in 24-well dishes containing 0.5 µg/mL of the respective chemokine (CCL17/CCL22 (Immunotools)) diluted in 500 μL assay medium. After 4 h of incubation at 37 °C and 5% CO_2_, migrated cells were quantified by measuring ATP using CellTiter Glo^®^ 2.0 (Promega) following the manufacturer’s protocol.

### 2.6. Flow Cytometry

For flow cytometry analysis, cells were collected, pelleted, and washed once with PBS. They were then resuspended with the appropriate volume of antibodies and incubated for 30 min at 4 °C. After antibody incubation, cells underwent two 5 min washes with PBS at 300× *g* before being resuspended in PBS and analyzed using a flow cytometer. The antibodies used included CD3 (clone HIT3b), CD8 (clone MEM-31), CD56 (clone BA19), and CD279 (clone EH12.2H7), provided by Immunotools. Anti-CEA/3 (clone 308/3-3), CD194/CCR4 (clone L291H4), Anti-CD94 (clone DXX22), CD274 (clone 29E.2A2), streptavidin PE, and AF647, as well as IgG Fc (clone QA19A42), were supplied by Biolegend (San Diego, CA, USA). Polyclonal anti-IgGFc was provided by Jackson Immunoresearch. Fluorescence measurements were conducted using a BD FACScalibur™, and the data were analyzed with FlowJo v10.8 (Becton, Dickinson and Company; Franklin Lakes, NJ, USA; 2019).

### 2.7. Immunocytochemistry and Confocal Microscopy

Multicellular tumor spheroids (MCTS) were subjected to a series of preparation steps. Initially, MCTS were collected and washed three times in PBS. Between each wash, the spheroids were allowed to settle at the bottom of the reaction tube, and the supernatant was aspirated. Following the washing steps, the MCTS were fixed in 4% paraformaldehyde for 20 min. Subsequently, they were blocked with a 3% BSA solution in PBS for 1 h at room temperature before they were left to react with the indicated antibodies for 1 h, or alternatively, overnight at 4 degrees Celsius. After the antibody incubation, the spheroids were mounted on microscope slides using ROTI^®^Mount FluorCare mounting solution. Confocal imaging was performed using a Leica TCS SPE confocal microscope (Leica, Wetzlar, Germany), and image analysis was conducted with Image J, version 1.37 (U.S. National Institutes of Health, Bethesda, MD, USA).

### 2.8. Trogocytosis Assays

Trogocytosis assays were performed as previously described in the star protocol from Delgado et al. with minor changes [34]. Briefly, 20,000 target cells (LS174T, SW1222, SW1417, HT-29, BJ1), expressing CEA were co-cultured with 20,000 CEA-CAR NK-92 cells per well for 4 h in a 96-well plate. After co-culture, three wells of the cells were pooled, collected, and stained for CD56 and CEA and directly analyzed in a flow cytometer. To decrease background staining and control for unspecific antibody labeling, the data were normalized against a CEA-negative cell line (BJ1). Inhibition of trogocytosis was performed by culturing the cells with 10 µM Cytochalasin 2 h before starting the trogocytosis assay.

Trogocytosis-induced fratricide was measured by letting CFDA-labeled CEA-CAR NK cells or untransduced NK-92 cells react with SW1222 cells as mentioned above. After 4 h of co-culture, 20,000 unlabeled effector cells were added to the co-culture and left to react for 18 h resulting in a 1:1:1 co-culture. Fratricide was determined by measuring the loss in CFDA-labeled population in the co-culture compared to control.

### 2.9. Data Analysis

Statistical analyses and graph presentations were conducted using Prism (v. 8.4.2, Graphpad). For normal distributed data, significance was determined using two-tailed unpaired *t*-test or multiple *t*-tests if not otherwise stated. For non-normally distributed data, the Mann–Whitney test was used to determine significance if not otherwise stated. Error bars are presented as the standard error of the mean (±SEM). Statistical significances are marked *p* < 0.05 = *, *p* < 0.01 = **, *p* < 0.001 = ***, *p* < 0.00001 = ****, and non-significant *p*-values are left unlabeled or marked ns.

## 3. Results

### 3.1. NK Cell CEA-CAR Validation 

We began by screening for CEA-binding antibodies and identified BW431/26, a CEA antibody that effectively binds membrane-bound CEA while remaining unaffected by soluble CEA forms [35,36,37]. This antibody, in its scFv form, was utilized as the binding region of the chimeric antigen receptor (CAR). Subsequently, a novel lentiviral vector with a CEA-specific CAR, a PD1x molecule, and a CCR4 receptor was engineered by fusing CEA-specific CAR domains with the hinge region of IgG1, the transmembrane and signaling domains of CD28, and the signaling domain of CD3ζ, and coupling this CAR to PD1x using P2A sequences (Figure 1). To promote tumor infiltration and enhance therapeutic efficacy, the CCR4 chemokine receptor was incorporated into the lentivirus vector. CCR4 induces chemotaxis in response to its target chemokines, CCL17 and CCL22, which are often upregulated in tumor microenvironments [38].

As has previously been described, we have identified and validated a naturally occurring splice variant of human PD1 checkpoint molecule (PD1x) that lacks the cytoplasmic signaling domain [39]. This PD1x variant acts as a competitive inhibitor to PD1 and also minimize off-target interactions of unwanted immune response. Additionally, PD1 is in general only expressed on highly activated NK cells, and in the case of some NK cell lines such as NK-92 cells, it is not detectable [40,41]. The expression of PD1x and more so CCR4 is weaker than the CAR element (Figure 1), which is typical for multigene constructs linked by 2A sequences [42].

We identified four CRC cell lines that presented significant (LS174T, SW1222, and SW1417) to moderate levels (HT-29) of CEA on the cell surface (Figure 2A,D) making those suitable targets to test the CEA-CAR. In order to validate the functionality of the CEA-CAR, we constructed Jurkat reporter cells which express GFP under the control of an NFkB-responsive promoter that activates when the CEA-CAR binds to the epitope of its target molecule (Figure 2B). As a control, we used Jurkat reporter cells with the same levels of CAR expression engineered with a control vector encoding for a non-binding CAR element identical to the CEA-CAR but without the binding region of the scFv fragment (Figure 2C). The CEA-CAR exhibited a strong response to LS174T and SW1222 which were the two cell lines expressing higher levels of CEA and no response to the CEA-negative cell line SKOV-3 (Figure 2E). Surprisingly, no activation could be seen for SW1417 and HT-29, despite decent levels of CEA expression (Figure 2D,E). The reduced activation might be due to differences between CAR format and antibody format [43], the epitope position within the CEA molecule directly impacting receptor-mediated CAR T-cell activation [44], or the role of adhesion molecules in efficient CAR recognition [45].

In order to test the cytotoxic capabilities of our CEA-CAR in vitro, we used two established NK cell lines, the NK-92 cell line, and the YT-cell line. Both cell lines were retrovirally transduced and maintained stable expression of both the CEA-CAR and the control CAR (Figure 3A). Next, the cell lines were challenged with cancer cells in a 2D cell culture setting.

The YT CEA-CAR cell line only showed significant killing capabilities against LS174T in the highest E:T ratio after 20 h of incubation (Figure 3B), while the NK-92 CEA-CAR showed significant levels of cytotoxic activity against LS174T, SW1222, and SW1417 and an elevated cytotoxic activity against HT-29 compared to the control CAR in a 4 h cytotoxicity assay (Figure 3C). No significant cytotoxic levels could be established against the CEA-negative BJ1 or MRC5 primary fibroblast cell lines, indicating that the CEA-CAR show low off-target effects (Figure 3C).

To assess and confirm the functionality of the PD1x splice variant in NK-92 cells, we first investigated the PD-L1 expression levels of the tested CRC cell lines and confirmed that the tested CRC cell lines all express PD-L1 (Figure 4A). Next, to determine potential synergistic or blocking effects of the PD1x splice variant on the CEA-CAR, we developed an assay including the checkpoint inhibitors, Pembrolizumab (PD1 blocker) and Atezolizumab (PD-L1 blocker), activated PBMCs, NK-92 cells, and NK-92-CEA-CAR cells and challenged them in different conditions against LS174T cells and SW1222 cells. The CEA-CAR cells exhibited a strong response against the CRC cell lines and were not influenced by the addition of PBMCs or checkpoint inhibitors (Figure 4B,C). This observation confirms that the PD1x splice variant acts as designed in not inhibiting the cytotoxic response of CEA-CAR-NK-92 cells against PD-L1-expressing cells.

To assess the functionality of the CCR4 receptor within the CAR construct, we conducted migration assays using CEA-CAR-transduced YT and NK-92 cells. These cells were exposed to the CCR4 receptor target chemokines CLL17 and CCL22 in a transwell insert system. Notably, both NK cell lines exhibited an augmented migratory response towards their respective target chemokines in comparison to untransduced control cells despite the low expression of surface CCR4 (Figure 4D).

In the case of the YT-cell line, a significant enhancement in migration towards CCL17 was observed, along with a heightened response to CCL22 (Figure 4E). This effect was further confirmed by comparing the internal migratory response of the untransduced control cells and the YT CEA-CAR cells in wells containing no chemokines (Figure 4E). In contrast, the NK-92 cell line demonstrated an elevated migratory response to the target chemokines (Figure 4F). However, compared to the control CAR, this response was not statistically significant, and this was also confirmed against the internal no chemokine control (Figure 4F). This discrepancy in response profiles between the two cell lines could arise from variations in the activation of their adhesion molecules and/or actin cytoskeletons, suggesting potential differences in intracellular signaling pathways. Alternatively, the expression profile of the receptors on the surface might be too low to induce a robust migratory response, consistent with previous reports [46].

Nevertheless, our experiments demonstrate the functionality of the CCR4 receptor within the CEA-CAR construct in its role of enhancing the migratory capabilities of these engineered immune cells.

### 3.2. Solid Tumor Modeling

In order to test the CEA-CAR in a model more true to in vivo conditions, we opted to create multicellular tumor spheroid models of the tested cell lines. This was carried out in two steps in order to build up complexity of the models. The first step was to observe growth and the ability to self-aggregate into solid shapes over time as monocultures. The second step was to increase the heterogeneity of the model by co-culturing cancer cells with fibroblasts which are known components of the tumor microenvironment [47].

After 4 days, all MCTS managed to form spherical solid shapes in the monocultures with a size of approximately 400 µm (Figure 5A,B); however, LS174T in monoculture MCTS formed a loose, flat, and unstable structure (Figure 5A). When co-cultured with fibroblasts, the LS174T: BJ MCTS formed compact structures that did not lose integrity when mechanically agitated, indicating that the added fibroblasts aided in forming a denser and tighter structure (Figure 5A). This relationship could not be established in the other cell line models and all other tested spheroids formed solid structures in the monocultures as well as in the co-culture setting.

When cultured on a low attachment surface, BJ fibroblasts form small and compact spheroid monocultures that do not change in size or metabolic activity (Figure 5D,E). However, in the case of MCTS co-cultures, no decrease in the growth (Figure 5D) or metabolic activity (Figure 5E) could be observed. Thus, the growth seen in the co-culture MCTS is due to cancer cell growth indicating that the fibroblasts act as a scaffold aiding the 3D structural formation of cancer cells in the co-culture MCTS. This observation is in line with previous reports utilizing co-culture MCTS [28,48]. Additionally, as can be seen for the case of LS174T:BJ and SW1222:BJ MCTS, the fibroblasts tend to co-localize in clusters in the middle of the spheroid (Figure 5C) where they probably act as a scaffold for the cancer cells giving them polarity through cell-to-cell and cell-to-extracellular matrix interactions.

Furthermore, we confirmed that the target antigen CEA and PD-L1 was displayed evenly and not masked on the surface of the MCTS due to the co-culture procedure making them suitable targets for the CEA-CAR against models with a characterized immunosuppressive phenotype (Figure 5F and Appendix A).

### 3.3. CEA-CAR NK Cells vs. Solid Tumor Models 

MCTS models have emerged as a more challenging and physiologically relevant platform for drug testing compared to standard 2D cell culture. This is attributed to the solid structure of MCTS, which mimics key features of the tumor microenvironment, such as physical barriers, nutrient deprivation, acidity, and hypoxia [28,49,50,51]. To recreate these conditions, we implemented an approach using tumor-conditioned media derived from MCTS. As a result, no medium changes were performed during the 4-day MCTS growth period, and the effector cells were directly added to the wells without washing away the tumor-conditioned media when performing cytotoxicity assays. This strategy combined with adding fibroblasts to the co-culture allowed us to more faithfully replicate the tumor microenvironment and investigate the response of effector cells within this complex milieu. Due to all MCTS being roughly the same size after 4 days, we could guarantee a standardized way of comparing the different E:T ratio effects of the CEA-CAR NK cells (Figure 5B). We first sought to validate the robustness of the CEA-CAR against MCTS models, as compared to a 2D setup by conducting a 4 h cytotoxicity assay using LS174T, LS174T:BJ, SW1222, and SW1222:BJ MCTS. Initial results revealed a minimal effect at the tested dosages prompting us to extend the assay time to 18 h to allow for a more comprehensive evaluation. Upon increasing the assay time, we observed significant cytotoxic responses against all the tested MCTS models except for the HT-29 MCTS models in which only a slightly elevated cytotoxic response compared to the control CAR was observed (Figure 6A). Notably, the cytotoxic effect was maintained to high levels, even in the co-culture models for all the tested MCTS models (Figure 6B). Furthermore, a higher activity of the control CAR cells can be seen in the MCTS models compared to the 2D data. This is especially prominent in the case of SW1222 and SW1417 in which the control CAR in the 2D 4 h assays yielded a minimal cytotoxic response, while in the 3D model it showed a high cytotoxic response, although not surpassing that of the CEA-CAR (Figure 3A and Figure 6A,B). The overall higher cytotoxic response observed in the MCTS models can be attributed to the longer incubation time, allowing for increased interaction between NK cells and the MCTS models, aided by the spatial distribution within the 3D structure. In the specific cases of SW1222 and SW1417 MCTS models, the heightened response may be attributed to intrinsic NK factors or higher expression of NK-specific ligands in the 3D structure compared to the 2D one that renders them particularly vulnerable to NK cell-mediated attack.

The tumor microenvironment is a multifaceted environment, and to further enhance its complexity, we introduced immune cells into our established MCTS models. Specifically, we chose to incorporate immune cells into the LS174T:BJ co-culture model due to its denser phenotype compared to its monoculture counterpart. We cultured activated PBMCs, primarily comprising of PD-L1-positve CD4 and CD8 T-cells, with the intention of simulating an immunologically inflamed tumor microenvironment (Figure 7A). These PBMCs were introduced 24 h after the initial spheroid formation, allowing them to effectively infiltrate and integrate into the model (Figure 7B). To guarantee a maintained phenotype of the PBMC, they were routinely checked before introduction into the MCTS (Figure 7C). Subsequently, the established triple-culture multicellular tumor spheroids were employed in cytotoxicity assays. Notably, the introduction of PBMCs into the MCTS model did not compromise the cytotoxicity of CEA-CAR NK-92 cells; they continued to exhibit robust cytotoxic activity when compared to PBMC alone (Figure 7D) and in a dose response fashion (Figure 7E). This observation underscores the compatibility of our model for investigating cytotoxicity within a complex TME as well as confirming the effectiveness of our CEA-CAR NK cells.

### 3.4. Trogocytosis and Trogocytosis-Induced Fratricide

Trogocytosis, the exchange of surface membrane molecules among immune cells, extends to interactions with both antigen-presenting cells and target cells of immune effector cells, influencing the presentation of these molecules on the cell surface. This phenomenon bears relevance to cell-based immunotherapies, with recent reports showing the role of chimeric antigen receptors (CAR) in trogocytosis-induced fratricide, an important limitation to CAR-NK cell expansion and efficacy [52,53].

Our findings show that trogocytosis indeed transpires between the CEA-CAR NK cells and the surface antigen of the target cell lines (Figure 8A). However, when we compared these findings to that of untransduced NK-92 cells, we found that the observed effect was not statistically significant (Figure 8B). The effect of trogocytosis could completely be blocked by Cytochalasin D which is a known compound inhibiting immunological synapse formation (Figure 8D). Taken together, this suggests that the trogocytosis observed is likely an intrinsic mechanism of NK-92 cells’ immunological synapse formation, rather than a consequence of the CEA-CAR. Given that the CEA-CAR is designed to recognize only the A3 epitope within the CEA molecule [44,54], we hypothesized that it might not respond to trogocytosis-acquired CEA molecules. To further investigate whether the trogocytosis observed in our samples might induce CAR-mediated fratricide, we designed an experiment with SW1222, the cell line with the highest observed trogocytosis in our experiments. In this assay, we allowed trogocytosis to occur between the target cells and labeled effector cells for 4 h followed by the addition of fresh unlabeled effector cells that were then left to react for 18 h (Figure 8C). After this co-incubation period, we examined the samples for any increased specific cell death in comparison to untransduced control cells. As anticipated, our results indicated that there was no discernible increase in cell death, indicating that our CEA-CAR does not exacerbate trogocytosis-induced fratricide when compared to our untransduced control cells in this setting (Figure 8E).

## 4. Discussion

Colon cancer stands as a prominent contributor to global mortality, with current treatments primarily revolving around surgical resection and adjuvant chemotherapy, with recurrence rates ranging from 5% to 30% [55]. Given the projected global surge in colorectal cancer incidence to 3.2 million cases by 2040 [56], and a dismal 5-year survival rate of 15–20% even in molecularly defined prognostic subgroups in the metastatic setting, there is an urgent demand for innovative, targeted treatment strategies.

CEA, a well-recognized biomarker for colorectal cancer, is a fetal glycoprotein that becomes elevated in CRC. Consequently, it has served as the focal point for various clinical trials and targeted therapies. Among these approaches, CEA-CAR therapy has been demonstrated to be well tolerated in clinical trials [57,58,59]. This paper introduces a new-generation NK cell CEA-CAR therapy designed for the targeted treatment of CEA-expressing colorectal cancer (CRC) and presents its efficacy against multiple in vitro models of colon carcinoma.

In solid tumor therapy, overcoming the tumor microenvironment is a significant challenge. Our CEA-CAR design aimed to address TME complexities by incorporating a PD1x splice variant, acting as a competitive inhibitor to disrupt endogenous PD1-PD1 ligand interactions within the TME, thereby enhancing tumor-specific responses. Additionally, we assessed the functionality of a CCR4 receptor, designed to enhance homing towards the tumor mass.

Functionality of the PD1x splice variant was extensively validated in previous work [39] and was further confirmed to not inhibit NK-92 cells transduced with the CEA-CAR. Its inclusion did not compromise the efficiency of our CEA-CAR cells against various in vitro models expressing an immune suppressive phenotype through PD-L1 in 2D and 3D models. Additionally, we could show that the addition of activated PBMC or the commonly used checkpoint inhibitors Atezolizumab and Pembrolizumab did not affect the CEA-CAR cells in producing a robust cytotoxic response against CEA-expressing target models. These findings further open up the potential for combination therapy approaches involving CEA-CAR cells and checkpoint inhibitors.

CCR4-mediated chemotaxis plays a pivotal role in the progression of various solid tumors, with its primary ligands being produced by T-regulatory cells and macrophages within the tumor microenvironment [38]. High CCR4 expression in colorectal cancer tissues has been associated with shorter overall survival and enhanced metastasis [60], emphasizing its relevance as a homing marker for targeted therapies. Our findings validate that the incorporated CCR4 element retained functionality despite its low membrane surface expression, likely a consequence of its position as the terminal element in the gene cassette. The response towards CCL17 and CCL22 chemokines could be further amplified by overexpressing the CCR4 receptor, as demonstrated in natural killer cells in previous studies [46]. However, this would necessitate a simpler genetic construct design, potentially omitting the PD1x component. Additionally, the utilized chemokine concentrations (0.5 µg/mL) may not accurately reflect physiological conditions, potentially saturating the reaction and influencing the migratory response in effector cells. Future research should focus on optimizing CCR4 receptor surface expression within the CEA-CAR construct and evaluating its effects on migration in in vivo models to validate its migratory-augmenting properties.

The utilization of MCTS models in drug testing has become increasingly common due to their ability to provide a more challenging and physiologically relevant representation of the tumor microenvironment, surpassing the limitations of traditional 2D cell culture in tumor modeling [32,33,49]. The solid structure of MCTS closely mimics key features of the tumor microenvironment, including physical barriers, nutrient depravation, acidity, and hypoxia [29,30,50].

In our study, we employed a strategy to accurately recreate these conditions by utilizing tumor-conditioned media derived from MCTS and by incorporating fibroblasts and additionally activated immune cells which are known tumor microenvironment components. Cancer-associated fibroblasts have been shown to aid the tumor in the tumor microenvironment by creating physical barriers through the production of a stiff ECM, growth-stimulating signals, and specifically, NK cell inhibitory molecules such as TGF-beta [23,47,61]. When tumor-conditioned media from different CRC cell lines were applied onto fibroblast cultures, they increased their TGF-beta production, further implicating the immunosuppressive effect of fibroblasts and the use of tumor-conditioned media in model systems [62]. Notably, CEA-CAR cells effectively eliminated multiple tested MCTS models even when incorporating fibroblasts, demonstrating their ability to overcome TME challenges. The role of immunologically “hot” tumors in cancer prognosis, including CRC, is complex and multifaceted, influenced by factors such as immune cell composition, patient health, and age [63,64,65]. While a high infiltration of immune cells in “hot” tumors generally correlates with positive prognostic outcomes, in some cases, chronic inflammation may enable tumor immune escape mechanisms [65,66,67]. Additionally, some reports suggest that heavy infiltration of cancer-associated fibroblasts (CAFs) combined with immune cell infiltration can lead to increased expression of immune checkpoint molecules, negatively impacting disease progression [68,69]. To simulate this complex milieu and assess its effect on CEA-CAR efficacy, we introduced activated PBMC to our established models. We observed no decrease in CEA-CAR efficiency; if anything, the addition of activated PBMC appeared to enhance the effectiveness of CEA-CAR cells. This aligns with previous research suggesting that higher counts of infiltrating lymphocytes lead to better responses in immunotherapy treatments [65]. Our results demonstrate the potency and efficacy of CEA-CAR in inducing cytotoxic responses against various MCTS models, highlighting its potential as a therapeutic strategy for treating colon carcinoma and other solid tumors expressing CEA.

Our findings demonstrate that trogocytosis, the exchange of surface membrane molecules between cells, occurs between CEA-CAR NK-92 cells and the surface antigen of target cell lines. This observation is consistent with previous reports indicating that trogocytosis is a widespread phenomenon among immune cells [70,71]. However, our data suggest that this trogocytosis is an intrinsic mechanism of NK-92 cells linked to NK receptor recognition and immunological synapse formation, rather than a consequence of CEA-CAR expression. This is supported by similar trogocytosis levels in untransduced and CEA-CAR NK-92 cells and by the inhibition of trogocytosis by compounds that block immunological synapse formation. Importantly, we observed no exacerbation of CAR-mediated fratricide by trogocytosis. This could be due to the low levels of trogocytosis observed in our experiments, or to the specificity of our CEA-CAR for membrane-bound CEA, which may prevent it from reacting to trogocytosis-acquired molecules. Further investigation of this hypothesis is warranted.

## 5. Conclusions

In conclusion, the utilization of MCTS co-culture models combined with the implementation of tumor-conditioned media and activated immune cells has provided a valuable tool for investigating the behavior of effector cells within a microenvironment that mimics solid tumors. The observed cytotoxic responses against MCTS models validate the potential of CEA-CAR in effectively targeting and eliminating tumor cells within this milieu. Additionally, our data suggest that our CEA-CAR does not cause CAR-mediated fratricide and that the remaining trogocytosis is an intrinsic mechanism of NK-92 cell immunological synapse formation. Further exploration of these findings may contribute to the development of more effective immunotherapeutic strategies against colon carcinoma and other CEA-positive malignancies. Future work should extend our findings in vivo before proceeding to clinical trial.

## Figures and Tables

**Figure 1 cancers-16-00388-f001:**
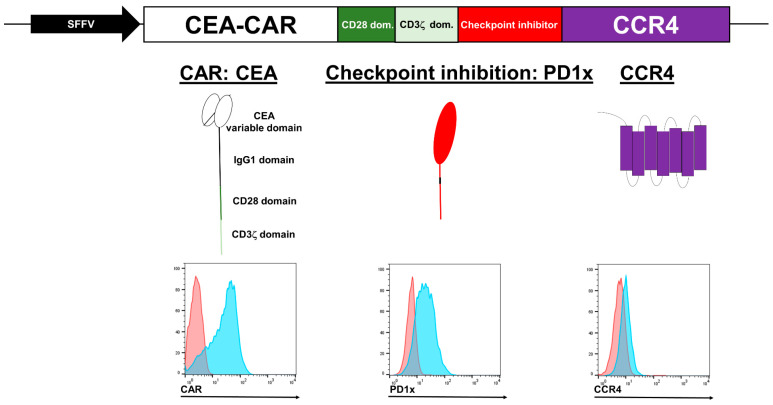
Schematic representation of the CEA-specific CAR lentivirus vector. The vector includes a checkpoint inhibition molecule (PD1x) designed to competitively inhibit PD1 and a CCR4 receptor designed to increase homing towards solid tumors. All elements are expressed under the control of a constitutive SSFV promoter. Flow cytometric representation of the general expression of each CAR element in transduced NK-92 cells is shown at the bottom of the depiction, red curves represent the control stain while blue curves represent the stain for the target molecule in the histogram.

**Figure 2 cancers-16-00388-f002:**
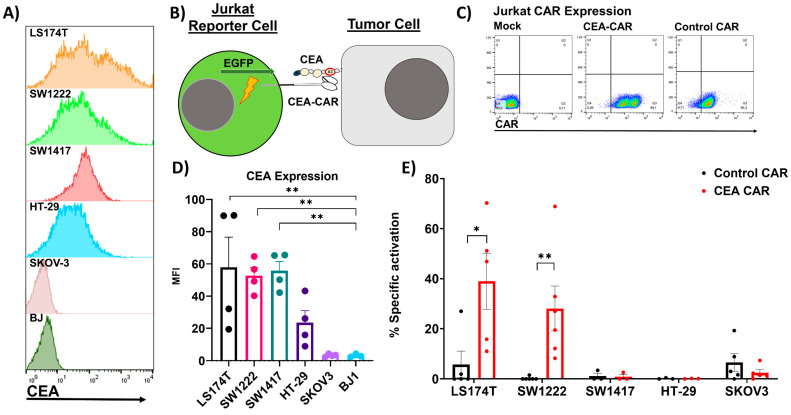
CEA expression on target cell lines and recognition by CEA-CAR: (**A**) Representative FACS data of CEA expression on colorectal carcinoma cell lines (LS174TT, SW1222, SW1417, and HT-29) alongside CEA-negative cell lines (SKOV-3 and BJ1) as a control. (**B**) Graphical depiction of Jurkat reporter cell activation upon recognition of target antigen CEA, more specifically, the A3 part of CEA which is the epitope of the CEA-CAR. (**C**) FACS dot plots display CAR expression on Jurkat reporter cell lines employed for target screening and functional CAR validation. (**D**) Flow cytometric evaluation and summary of CEA expression on the assessed target cell lines. Expression levels are quantified as mean fluorescent intensity (MFI) derived from n = 4 biological replicates. Statistical significance was determined by one-way ANOVA. (**E**) The functionality of our construct was confirmed by employing Jurkat reporter cells encoding the EGFP gene under the control of an NFkB promoter. Jurkat reporter cells were co-cultured with target cell lines in 96-well plates for 18 h, at an effector-to-target (E:T) ratio of 1:1. The resulting increase in GFP+ Jurkat reporter cells following exposure to target cells or specific stimuli served as a measure of CAR stimulation. The results are expressed as a percentage of the maximum stimulus induced by PHA (1 µg/mL). This assay was performed with n = 3–5 biological replicates, each carried out in triplicate. Statistical significances are marked *p* < 0.05 = *, *p* < 0.01 = **, and non-significant *p*-values are left unlabeled or marked ns.

**Figure 3 cancers-16-00388-f003:**
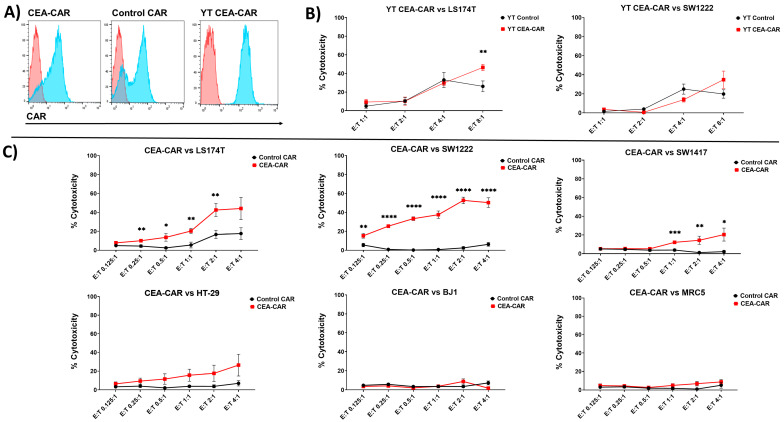
Efficient elimination of CRC cell lines by CEA-CAR in 2D cell culture: (**A**) Flow cytometric representation of the CAR expression for the CEA-CAR in NK-92 cells and YT-cells, red curves represent the control stain while blue curves represent CAR staining in the histogram. (**B**) Cytotoxicity data comparing CEA-CAR transduced YT-cells (red) against non-transduced control cells (black) when reacting with LS174T and SW1222 cell lines over an 18 h period at different E:T ratios. The graph data are obtained from n = 3 independent experiments, each conducted in triplicate. Significance was determined through unpaired multiple *t*-tests. (**C**) CEA-CAR efficiently eliminates CRC cell lines when co-cultured with colon carcinoma cell lines or non-cancerous cell lines for 4 h at varying effector-to-target (E:T) ratios. CEA-CAR cytotoxicity is represented by the red lines, while control CAR lacking the CEA recognition domain is depicted by the black lines. The graphs display data derived from n = 3 independent experiments, each conducted in triplicate. Statistical significance was assessed using unpaired multiple *t*-tests. Statistical significances are marked *p* < 0.05 = *, *p* < 0.01 = **, *p* < 0.001 = ***, *p* < 0.00001 = ****, and non-significant *p*-values are left unlabeled or marked ns.

**Figure 4 cancers-16-00388-f004:**
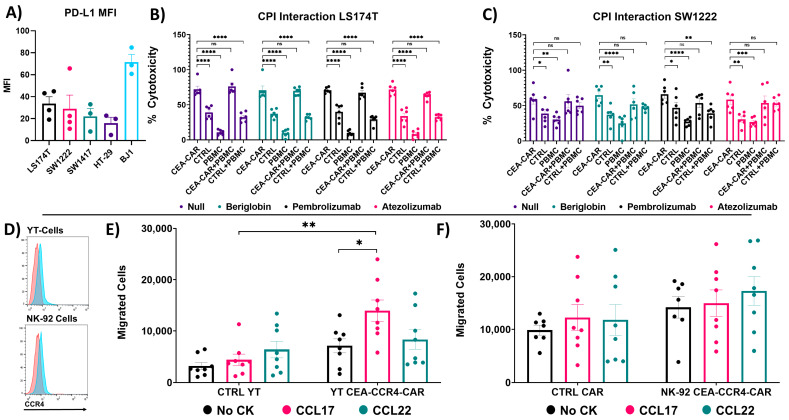
Functionality assessment of PD1x splice variant and CCR4 receptor: (**A**) Flow cytometric summary of PD-L1 expression on the assessed target cell lines. Expression levels are quantified as mean fluorescent intensity (MFI) derived from n = 4 biological replicates. (**B**) Assessment of PD1x splice variant and checkpoint inhibitor (CPI) interaction. Graphs represent 18 h cytotoxicity assays against LS174T cells at an effector target ratio 1:1 (5000:5000 cells) in the solo conditions and 1:1:1 (NK92: PBMC: LS174T) in the conditions where NK-92 cells and activated PBMC cells were employed. Pembrolizumab and Atezolizumab were added at a concentration of 10 µg/mL. Data represent n = 2 independent experiments performed in triplicate. Statistical significance was determined with a multiple comparison ANOVA with Holm–Sidak correction. (**C**) Similarly, the PD1x component of the CEA-CAR vector and CPI were assessed against the CRC cell line SW1222 with the same setup as in (**B**). Data represent n = 2 independent experiments performed in triplicate. Statistical significance was determined with a multiple comparison ANOVA with Holm–Sidak correction. (**D**) Flow cytometric analysis in the panels demonstrates the general CCR4 expression in both YT-cells and NK-92 cell lines, utilized for functionality testing of the CEA-CAR construct. The red curves represent the control stain while blue curves represent the CCR4 stain. (**E**) To assess the functionality of the CCR4 receptor in transduced effector cells, a transwell migration experiment was conducted, challenging the cells to migrate towards CCR4 receptor ligands, CCL17 and CCL22, through 8 µm thick transwell inserts in a 4 h assay. CEA-CAR YT-cells were evaluated against untransduced YT control cells to determine their migratory capabilities. The analysis includes the internal control migration of untransduced YT-cells and YT CEA-CCR4-CAR cells in response to no chemokines (CK). This analysis was performed across three independent experiments (n = 3), with each data point representing an individual transwell insert. Statistical significance was determined using an unpaired *t*-test. (**F**) Similarly, CEA-CAR NK-92 cells were compared to transduced control CAR cells lacking the CCR4 receptor for their migratory abilities toward CCR4 target chemokines in a transwell system. This assessment was repeated across three independent experiments (n = 3), with each data point representing an individual transwell insert. Statistical significance was determined using an unpaired *t*-test. Statistical significances are marked *p* < 0.05 = *, *p* < 0.01 = **, *p* < 0.001 = ***, *p* < 0.00001 = ****, and non-significant *p*-values are left unlabeled or marked ns.

**Figure 5 cancers-16-00388-f005:**
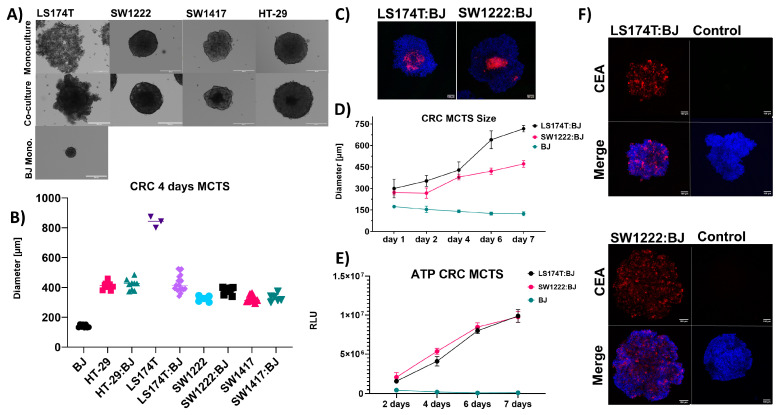
Generation and characterization of 3D CRC MCTS models: CRC target cell lines were tested and characterized for their ability to form solid structures as 3D tumor models in an attempt to mimic solid tumors. (**A**) Evaluation of the ability of CRC target cell lines to form 3D tumor models simulating solid tumors, presented as imaging of their general shape and structures in both monoculture and co-culture settings (CRC cell lines with fibroblasts). Scale bars are indicative of 250 µm. (**B**) A summary of the sizes of all tested MCTS models, both in monoculture and co-culture, following 4 days of incubation. Each data point represents the size of an individual MCTS. Assessments were conducted over n = 3–5 independent experiments. (**C**) Visualization of fibroblast co-localization within MCTS models in a co-culture configuration using LS17T4 and SW1222 cells. Fibroblast cell line BJ1, transduced with RFP, was detected via confocal microscopy. Images were captured at 10× magnification, with MCTS models counterstained with DAPI. Scale bars represent 100 µm. (**D**) A comparative analysis of the growth dynamics between two co-culture MCTS models (LS174T:BJ and SW1222:BJ) and monoculture spheroid aggregates of BJ1. Graphs illustrate size changes over time, with each data point originating from three independent experiments, n = 3, each with at least three technical replicates. (**E**) A comparative analysis of the metabolic activity between two co-culture MCTS models (LS174T:BJ and SW1222:BJ) and monoculture spheroid aggregates of BJ1. Graphs illustrate metabolic activity change over time measured through total ATP amount characterized through RLU (relative light unit), with each data point originating from three independent experiments (n = 3), each with at least three technical replicates. (**F**) CRC co-culture MCTS, approximately 400 µm in diameter, stained for the CAR target antigen CEA on the surface and counterstained with DAPI. Imaging was carried out using a confocal microscope at 10× magnification, excited with a solid-state laser (405 and 635 nm), with images presented as 3D projections of system-optimized z-stack. Control MCTS are stained with an isotype antibody to account for background staining.

**Figure 6 cancers-16-00388-f006:**
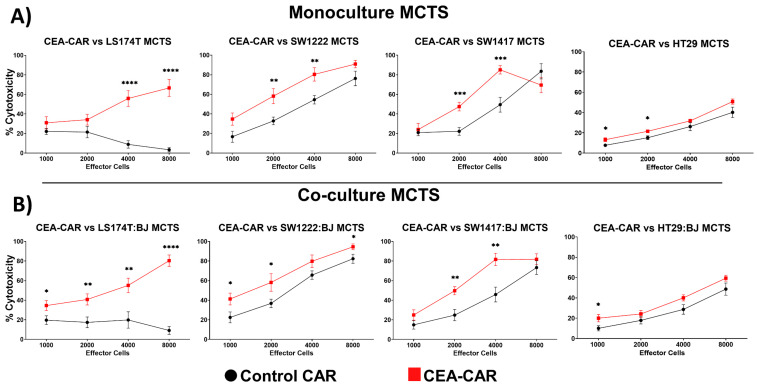
Effective elimination of solid tumor models by CEA-CAR: (**A**) CEA-CAR-transduced NK-92 cells demonstrate cytotoxic activity against monoculture MCTS solid tumor models at various effector-to-target (E:T) ratios shown as cell number employed against a single MCTS in that setting. Each data point represents results from n = 3 independently conducted experiments, each performed in triplicate. Statistical significance was determined using multiple *t*-tests. (**B**) Cytotoxic activity of CEA-CAR NK-92 cells against co-culture MCTS solid tumor models is displayed. Each data point represents outcomes from n = 3 independently performed experiments, each conducted in triplicate. Statistical significance was assessed using multiple *t*-tests. Statistical significances are marked *p* < 0.05 = *, *p* < 0.01 = **, *p* < 0.001 = ***, *p* < 0.00001 = ****, and non-significant *p*-values are left unlabeled or marked ns.

**Figure 7 cancers-16-00388-f007:**
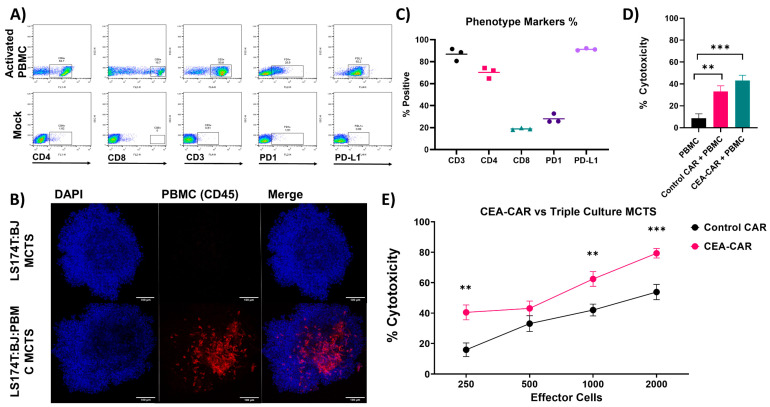
Effective eradication of triple culture CRC MCTS by CEA-CAR: (**A**) Flow cytometric representation of phenotypic markers of activated PBMC employed in the triple culture MCTS. Mock flow cytometric stains were conducted using isotype controls of the antibodies. (**B**) Imaging of triple culture MCTS, combining LS174T:BJ with activated leukocytes, captured by confocal microscopy at 20× magnification. Presented as a 3D projection of system-optimized z-stacks, activated leukocytes were detected using the pan leukocyte marker CD45. (**C**) The graph displays the percentage of positivity for the measured phenotypic markers. Each point corresponds to an independently performed experiment before integration into the triple culture MCTS. (**D**) To control for cytotoxicity effects from the activated PBMC population, an experiment was performed in which 500 cells of each cell type were employed against the triple culture MCTS in an 18 h assay. Data represent n = 2 independently conducted experiments performed in at least triplicate. Statistical significance was determined using a Mann–Whitney *t*-test. (**E**) CEA-CAR demonstrates efficacy in eliminating triple culture MCTS, even with the integration of immune cell components. The graph illustrates the cytotoxic activity of CEA-CAR compared to control CAR in an 18 h assay, with each data point representing results from n = 3 independently conducted experiments, each performed in triplicate. Statistical significance was assessed using multiple *t*-tests. Statistical significances are marked *p* < 0.01 = **, *p* < 0.001 = ***, and non-significant *p*-values are left unlabeled or marked ns.

**Figure 8 cancers-16-00388-f008:**
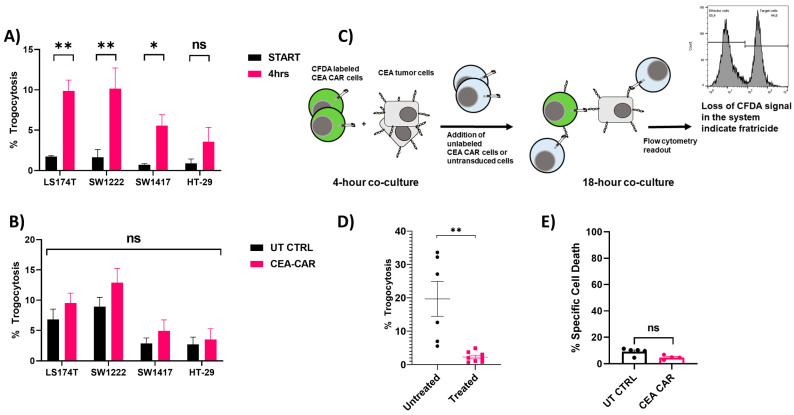
Trogocytosis of CEA-CAR and evaluation of CAR-mediated trogocytosis-induced fratricide: (**A**) Trogocytosis assessment involved NK-92 CEA-CAR cells interacting with different CEA-expressing target cell lines for 4 h, followed by flow cytometric evaluation of CEA on NK-92 CEA-CAR cells. The graph illustrates the percentage of positive CEA on NK-92 CEA-CAR cells in the sample at the assay’s outset (black) and after 4 h of co-incubation with target cell lines (red). Increased CEA positivity post co-incubation was characterized as % Trogocytosis. Data for each cell line are derived from n = 3 experiments, each conducted in triplicate. Statistical significance was determined using an unpaired *t*-test. (**B**) Trogocytosis assay measuring the % trogocytosis after 4 h of co-incubation with various target cell lines, comparing untransduced NK-92 cells (black) with CEA-CAR-transduced NK-92 cells (red). Data are collected from n = 4 different experiments, each performed in triplicate. Significance was tested using an unpaired *t*-test. (**C**) A graphical representation of the experimental setup used to determine CAR-mediated trogocytosis-induced fratricide. (**D**) Trogocytosis was effectively blocked by Cytochalasin C treatment, using the SW1222 cell line as the target cell line. Each point represents the mean of a triplicate and is derived from n = 3 independently conducted experiments. Statistical significance was determined by an unpaired *t*-test. (**E**) Measurement of trogocytosis-induced fratricide entailed allowing CFDA-labeled CEA-CAR NK cells or CFDA-labeled untransduced NK-92 cells to interact with SW1222 cells, followed by the addition of unlabeled CEA-CAR cells or unlabeled, untransduced NK-92 cells. The loss of the CFDA-labeled population in the co-culture was determined as % specific cell death, interpreted as Fratricide. The graph displays data from n = 2 independently conducted experiments with each point representing the mean of 96-well plate triplicates, with significance assessed using the Mann–Whitney *t*-test. Statistical significances are marked *p* < 0.05 = *, *p* < 0.01 = **, and non-significant p-values are left unlabeled or marked ns.

## Data Availability

The data could be given upon reasonable request from the corresponding author.

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
