# Peer review of "Next-Generation CEA-CAR-NK-92 Cells against Solid Tumors: Overcoming Tumor Microenvironment Challenges in Colorectal Cancer"

_cancers, 2024, doi:10.3390/cancers16020388_

Round 1
Reviewer 1 Report
Comments and Suggestions for Authors
1) In this study, cancer cell lines such as NK92 and YT cells were utilized instead of NK cells isolated from human blood. Since this CAR (Chimeric Antigen Receptor) is based on cancer cell lines, it cannot be claimed to be entirely free from risks. Therefore, it is required to replicate the experiments using NK cells derived from humans (patients).
2)The authors of this paper performed organoid experiments as an alternative to animal experiments. Although organoids can mimic actual organ functions and enable the confirmation of effectiveness without the need for animal experiments, they fall short of replicating the interactions among organs that can be observed in treatments based on animal experiments.
Comments on the Quality of English Language
Editing of English language required.
Author Response
Reviewer 1 comments:
In this study, cancer cell lines such as NK92 and YT cells were utilized instead of NK cells isolated from human blood. Since this CAR (Chimeric Antigen Receptor) is based on cancer cell lines, it cannot be claimed to be entirely free from risks. Therefore, it is required to replicate the experiments using NK cells derived from humans (patients).
- We recognize the reviewers comment and we share the opinion. However, this paper was focused on looking at NK cell lines such as the NK-92 cell line as an alternative source of NK cells for CAR therapy due to their advantages in expandability and production. NK92 in particular has been already used in several clinical trials. The methods used in the paper are theoretically transferable to a primary NK platform in which patient or healthy donor NKs are used. This is however outside the scope of this paper.
The authors of this paper performed organoid experiments as an alternative to animal experiments. Although organoids can mimic actual organ functions and enable the confirmation of effectiveness without the need for animal experiments, they fall short of replicating the interactions among organs that can be observed in treatments based on animal experiments.
- We recognize the reviewers comment and agree that a shortcoming of the paper is that we did not replicate and directly compare our findings in an animal model. However, we aimed to establish multiple in vitro model systems that would mimic components of the tumor microenvironment without the need for an animal model. As the reviewer points out, the interaction between other organs cannot be established, however, the aim of the paper was to model and overcome tumor microenvironment components and not to investigate interactions of the treatment with other organs. Toxicity studies and potential in vivo interactions of the CAR therapy in animal models will be the focus of future testing.
Reviewer 2 Report
Comments and Suggestions for Authors
In this study Franzen and coworkers describe a novel CEA-CAR construct transduced in NK92 and YT cell lines. They validate CEA-CAR NK92 in 2D and 3D models showing good efficacy of their CAR model also excluding a relevant role of a fratricide effect linked to CEA trogocytosis by effector cells.
The novel CAR construct integrating PD1x proposed by the authors as well as the use of 3D models also including fibroblasts cell lines to assess CAR NK92 activity are strenghts of this work.
On the other hand the presence of CCR4 in the construct which is potentially very interesting, has not a significant impact on CAR-NK92 while the impact of PD1x has not been addressed here. Moreover the choice of NK92-CAR platform brings concerns in view of a clinical translation.
The YT model would be better excluded from the article since it has not been validated in 3d models and does not offer advantages in 2d models compared to NK92. Indeed YT are known to be to be poorly cytotoxic.
Some experimental controls and direct comparisons of different conditions are missing
Major revisions
-The impact of PD1x should be addressed in this experimental system, otherwise considering the low impact of CCR4, the novelty of this car product is limited.
-In Fig 4 I would let only C and D panels adding the appropriate stat significance and the CCR 4 expression profiles
-In fig. 6 a comparison in the same experiment between CEA CAR NK92-mediated killing of monoculture vs coculture is missing.
-Line 408 Besides intrinsic NK factors, the higher susceptibility of SW1222 could be related to a different expression of ligands for NK receptors in the 3D vs 2D models (higher for act NK and/or lower for inhibitory rec?). This aspect could be important to further confirm the relevance of the 3D model proposed.
The SW1417 3D model is pretty different compared to 2D. It should be described as well
-In fig7 panel B the experiment must include a direct comparison between triple culture and coculture with statistical significance. ActPBMC alone without CEA CAR NK should be included as control. Indeed comparing fig 6b with 7b it appears that cytotoxicity in triple culture is much higher than in coculture. This increase could be explained by actPBMC cytotoxic properties vs LS174T:BJ. Act PBMC could also include actNK cells likely cytotoxic vs Tumor cell lines
-lines 541-542. The authors claim they have used a conditoined media in their system. This was not clearly reported in materials and methods or in results or fig legends
Minor comments
-LINE135 presence not present
-Line 382: deprivation instead of depravity
-In Fig 5 panel D and E legends are the same. In panel E y-axis it’s reported RLU. Is it really that or is it the cell numbers? If possible it would be clearer if cell numbers instead of RLU were reported
-The panels order in fig 7 should follow the description in the text. Panel b could be moved to supplemental material
-Comments on other 3D models commonly employed that include ECM-mimicking materials would be useful in discussion-
-the CTRL CAR construct could be better explained for those readers not expert in the field
- comments on other CEA CARNK92 constructs could be added in introduction or discussion
Comments on the Quality of English LanguageWell written, minor mispelling errors
Author Response
Reviewer 2 comments:
In this study Franzen and coworkers describe a novel CEA-CAR construct transduced in NK92 and YT cell lines. They validate CEA-CAR NK92 in 2D and 3D models showing good efficacy of their CAR model also excluding a relevant role of a fratricide effect linked to CEA trogocytosis by effector cells.
The novel CAR construct integrating PD1x proposed by the authors as well as the use of 3D models also including fibroblasts cell lines to assess CAR NK92 activity are strenghts of this work.
On the other hand the presence of CCR4 in the construct which is potentially very interesting, has not a significant impact on CAR-NK92 while the impact of PD1x has not been addressed here. Moreover the choice of NK92-CAR platform brings concerns in view of a clinical translation.
The YT model would be better excluded from the article since it has not been validated in 3d models and does not offer advantages in 2d models compared to NK92. Indeed YT are known to be to be poorly cytotoxic.
Some experimental controls and direct comparisons of different conditions are missing
Major revisions
The impact of PD1x should be addressed in this experimental system, otherwise considering the low impact of CCR4, the novelty of this car product is limited.
- We have added an experiment investigating the PD1x functionality and CPI interaction (Figure 4B,4C) and expanded further upon this in the results and discussion. (Line 318-329, 556-564)
In Fig 4 I would let only C and D panels adding the appropriate stat significance and the CCR 4 expression profiles
- We thank the reviewer for the suggestion and agree that a more compact presentation of the data is more suitable and have amended the figure correspondingly.
In fig. 6 a comparison in the same experiment between CEA CAR NK92-mediated killing of monoculture vs coculture is missing.
- The experiments shown are done in parallel with the same effector cells and target MCTS with the difference of the mono and the co-culture settings. As we found no significant difference between monoculture and co-culture we show both datasets separately for clarity.
Line 408 Besides intrinsic NK factors, the higher susceptibility of SW1222 could be related to a different expression of ligands for NK receptors in the 3D vs 2D models (higher for act NK and/or lower for inhibitory rec?). This aspect could be important to further confirm the relevance of the 3D model proposed.
- We thank the reviewer for the suggestion and have expanded upon this in the text. ( Line 441-444). We will look into it in future research to confirm if MCTS (3D) might have higher NK specific receptors compared to 2D culture.
The SW1417 3D model is pretty different compared to 2D. It should be described as well
- We have now expanded the SW1417 results in the text. (Line 435-444)
-In fig7 panel B the experiment must include a direct comparison between triple culture and coculture with statistical significance. ActPBMC alone without CEA CAR NK should be included as control. Indeed comparing fig 6b with 7b it appears that cytotoxicity in triple culture is much higher than in coculture. This increase could be explained by actPBMC cytotoxic properties vs LS174T:BJ. Act PBMC could also include actNK cells likely cytotoxic vs Tumor cell lines.
- We left out the PBMC control as we could show that they were ineffective alone towards the MCTS. Additionally, routine screening showed that the CD56-expressing cells in PBMCs were below 10% in line with the phenotype data having roughly 90% CD3 positive cells. We have added the data with a comparison of PBMC alone vs NK92 with PBMCs showing that PBMC alone does not have any significant cytotoxic effect towards the triple culture model, however when combined with NK92 cells the cytotoxic effect is significantly enhanced confirming that the bulk cytotoxic effect seen can be coupled to the addition of CEA-NK92 cells.
lines 541-542. The authors claim they have used a conditoined media in their system. This was not clearly reported in materials and methods or in results or fig legends
- We have now addressed it in the material and methods (line 136-138), and in the results part (line 416-423).
Minor comments
LINE135 presence not present
- fixed
Line 382: deprivation instead of depravity
- fixed
In Fig 5 panel D and E legends are the same. In panel E y-axis it’s reported RLU. Is it really that or is it the cell numbers? If possible it would be clearer if cell numbers instead of RLU were reported
- We thank the reviewer for noticing this error in the description of figure 5 E. In the figure, we are analyzing the RLU representing ATP as a means of metabolic activity of the cells over time. Point being that the cells not only keeps growing in size (figure 5 D) they also maintain their metabolic activity over time, which also could be quantified to cell numbers.
The panels order in fig 7 should follow the description in the text. Panel b could be moved to supplemental material
- We agree with the reviewer and have fixed the order of the panels in figure 7.
Comments on other 3D models commonly employed that include ECM-mimicking materials would be useful in discussion.
- We agree with the reviewer that it would be interesting to include other 3D models with ECM mimicking materials such as 3D-printed Organoids, Matrigel domes, Layered cell cultures or even microfluidic chip technologies in the discussion but due to conciseness, we decided that it is outside the scope of the paper and left it out.
the CTRL CAR construct could be better explained for those readers not expert in the field
- We have more thoroughly explained the control vector used in the text. ( Line 242-245)
Reviewer 3 Report
Comments and Suggestions for Authors
This manuscript introduces a novel NK cell CEA-CAR therapy for colorectal cancer (CRC). This innovative approach incorporates a PD1x splice variant to disrupt tumor microenvironment (TME) interactions and a CCR4 receptor for enhanced tumor homing. The research successfully demonstrates the efficacy of CEA-CAR cells against various in vitro models of colon carcinoma, overcoming TME challenges in multicellular tumor spheroids. Additionally, the study explores trogocytosis as an intrinsic NK-92 cell mechanism unrelated to CAR expression, with no observed exacerbation of CAR-mediated fratricide. These findings, emphasizing the efficacy and potential of CEA-CAR therapy, provide valuable insights into immunotherapeutic strategies against CRC and other CEA-positive malignancies. While the manuscript presents a promising approach with notable strengths, considerations for the followings could strengthen its translational potential.
1. Conduct in vivo experiments to validate the efficacy of CEA-CAR cells in animal models.
2. Explore potential combination therapies that could enhance the effectiveness of CEA-CAR cells by combing existing treatments, such as chemotherapy or immune checkpoint inhibitors to check potential synergistic effects. Additionally, a direct comparison of CEA-CAR therapy with existing standard treatments would be ideal.
3. Investigate the impact of an immunosuppressive microenvironment on CEA-CAR efficacy. For example, conditions with high levels of cancer-associated fibroblasts and immune checkpoint molecules.
4. Any safety and potential toxicities associated with CEA-CAR therapy?
Author Response
Reviewer comments 3:
This manuscript introduces a novel NK cell CEA-CAR therapy for colorectal cancer (CRC). This innovative approach incorporates a PD1x splice variant to disrupt tumor microenvironment (TME) interactions and a CCR4 receptor for enhanced tumor homing. The research successfully demonstrates the efficacy of CEA-CAR cells against various in vitro models of colon carcinoma, overcoming TME challenges in multicellular tumor spheroids. Additionally, the study explores trogocytosis as an intrinsic NK-92 cell mechanism unrelated to CAR expression, with no observed exacerbation of CAR-mediated fratricide. These findings, emphasizing the efficacy and potential of CEA-CAR therapy, provide valuable insights into immunotherapeutic strategies against CRC and other CEA-positive malignancies. While the manuscript presents a promising approach with notable strengths, considerations for the followings could strengthen its translational potential.
- Conduct in vivo experiments to validate the efficacy of CEA-CAR cells in animal models.
- We recognize the reviewers comment and agree that a shortcoming of the paper is that we did not replicate and directly compare our findings in an animal model. Toxicity studies and potential in vivo interactions of the CAR therapy in animal models will be the focus of future testing.
- Explore potential combination therapies that could enhance the effectiveness of CEA-CAR cells by combing existing treatments, such as chemotherapy or immune checkpoint inhibitors to check potential synergistic effects. Additionally, a direct comparison of CEA-CAR therapy with existing standard treatments would be ideal.
- We have now added a CPI experiment to investigate this (Figure 4).
- Investigate the impact of an immunosuppressive microenvironment on CEA-CAR efficacy. For example, conditions with high levels of cancer-associated fibroblasts and immune checkpoint molecules.
- PD-L1 is naturally occurring in many of the tested cell lines; we have included additional analyses of PD-L1 in 2D and 3D data (Figure 4, Figure S1) and expanded upon this further in the paper. (Line 300-310, Line 408-411)
- Any safety and potential toxicities associated with CEA-CAR therapy?
- Previous clinical studies looking at CEA-CAR therapy have reported that CEA-CAR therapy in general is well tolerated with no severe adverse effects related to the CEA-CAR therapy. (DOI: 10.1016/j.ymthe.2017.03.010)
- We have expanded upon this further and made it clearer in the discussion. (line 543-546).
Reviewer 4 Report
Comments and Suggestions for Authors
In this manuscript, Franzén and colleagues developed a CEA-CAR-NK-92 anti tumor cell line engineered with a new vector transfecting PD-1 and CCR4 in addition to CAR-CEA and they explored its effectiveness in 2D and 3D culture models against CRC cell lines. Leaving out the possible role of PD1 that the authors explicitly do not take into consideration in this work (and whose function it’s already analyzed in previous paper -ref 38), CEA-CAR-NK-92 cell line has shown to almost always exert a higher cytotoxicity against CEA positive CRC cell lines (in both 2D and 3D models) but it does not show increased migration capability in response to CCL17/CCL22. Of sure, CEA-CAR mediated CRC killing well works and data obtained in 3D co-culture MCTS suggest that its efficiency could not be strongly affected by TME, even if there is no characterization of tumor conditioned media. However the low expression of CCR4 on CEA-CAR-NK-92 and the lack of a significant migration driven by CCL17/CCL22 could affect the novelty and the impact of this new engineered cells for future application for immunotherapy. These also in consideration of platform used for CAR transfection: indeed NK cell lines presents issues in immunotherapy that needs to be cited at least in the introduction paragraph (see below point 1).
In the last part of manuscript authors evaluate trogocytosis of CEA-CAR and suggested that the expression of this new receptor does not increase the exchange of surface molecules between effector and target cells such that a fratricidal effect could be generated. These data are of interest since the fratricide effect can represent a limitation to CAR-NK efficacy.
Here, some requests in detail:
1. Introduction
a) lines 53-61. Some additional hurdles in the use of NK-92 as source for CAR products should be cited, such as the fact that NK-92 cells are aneuploid and of malignant origin, thus requiring irradiation before cell infusion and that irradiation can limit the persistence of NK-92 cells and negatively impact their durable therapeutic efficacy. In addition, NK-92 are naturally deprived of CD16, indicating a deficiency in the ADCC response, useful for possible mAb combined immunotherapies. (Zhang, Y., Zhou, W., Yang, J. et al. Chimeric antigen receptor engineered natural killer cells for cancer therapy. Exp Hematol Oncol 12, 70 (2023). https://doi.org/10.1186/s40164-023-00431-0)
2. Material and Methods
a) Cell lines and Cell Culture:
BJ1 are not mentioned, please add them, their histotype and their culture medium in the paragraph
b) Cytotoxicity Assay:
Pls, explain more in detail the principle of the method that is usually used for cell viability assay more than cytotoxicity assay.
c) Immunocytochemistry and confocal microscopy:
Line 166: pls add abbreviation “MCTS” after multicellular tumor spheroids
d) Trogocytosis Assay
Pls start the paragraph with a subject
3. Results:
a) Pls number the figures always following the order of citation in the main text and also change the order of the panels in each figure based on what is cited first in the manuscript
b) Pls better specify what is meant by Control CAR (paragraph 3.1)
c) Fig.4: histogram graph in panel A is redundant to panel C as well as histogram graph in panel B to panel D. I suggest to display only graphs C and D.
d) Fig. 7: these data should be supported by other experimental results. I suggest the authors to add citotoxicity results of triple culture MCTS with not activated PBMCs to be performed and analyzed in comparison to PHA activated PBMC. Indeed the not activated PBMC could be a better resembling condition of in vivo status and data obtained from these experiments have to be compared to data obtained with PHA activated PBMC. Indeed, the higher cytotoxic activity, shown in panel B (i.e. CEA-CAR vs LS174T:BJ in fig 7B vs fig 6B), could be simply due a killing activity of effector cells present in activated PBMCs (add in panel D also the % of NK cells) or to secreted factors, cell contacts mediated by activated blood derived cells obtained from an in vitro setting difficult to reproduce in vivo.
Author Response
Reviewer comments 4:
In this manuscript, Franzén and colleagues developed a CEA-CAR-NK-92 anti tumor cell line engineered with a new vector transfecting PD-1 and CCR4 in addition to CAR-CEA and they explored its effectiveness in 2D and 3D culture models against CRC cell lines. Leaving out the possible role of PD1 that the authors explicitly do not take into consideration in this work (and whose function it’s already analyzed in previous paper -ref 38), CEA-CAR-NK-92 cell line has shown to almost always exert a higher cytotoxicity against CEA positive CRC cell lines (in both 2D and 3D models) but it does not show increased migration capability in response to CCL17/CCL22. Of sure, CEA-CAR mediated CRC killing well works and data obtained in 3D co-culture MCTS suggest that its efficiency could not be strongly affected by TME, even if there is no characterization of tumor conditioned media. However the low expression of CCR4 on CEA-CAR-NK-92 and the lack of a significant migration driven by CCL17/CCL22 could affect the novelty and the impact of this new engineered cells for future application for immunotherapy. These also in consideration of platform used for CAR transfection: indeed NK cell lines presents issues in immunotherapy that needs to be cited at least in the introduction paragraph (see below point 1).
In the last part of manuscript authors evaluate trogocytosis of CEA-CAR and suggested that the expression of this new receptor does not increase the exchange of surface molecules between effector and target cells such that a fratricidal effect could be generated. These data are of interest since the fratricide effect can represent a limitation to CAR-NK efficacy.
Here, some requests in detail:
- Introduction
- a) lines 53-61. Some additional hurdles in the use of NK-92 as source for CAR products should be cited, such as the fact that NK-92 cells are aneuploid and of malignant origin, thus requiring irradiation before cell infusion and that irradiation can limit the persistence of NK-92 cells and negatively impact their durable therapeutic efficacy. In addition, NK-92 are naturally deprived of CD16, indicating a deficiency in the ADCC response, useful for possible mAb combined immunotherapies. (Zhang, Y., Zhou, W., Yang, J. et al. Chimeric antigen receptor engineered natural killer cells for cancer therapy. Exp Hematol Oncol 12, 70 (2023). https://doi.org/10.1186/s40164-023-00431-0)
- We have expanded upon this further in the text. (line 69-74)
- Material and Methods
- a) Cell lines and Cell Culture:
BJ1 are not mentioned, please add them, their histotype and their culture medium in the paragraph
- fixed
- b) Cytotoxicity Assay:
Pls, explain more in detail the principle of the method that is usually used for cell viability assay more than cytotoxicity assay.
- We have further expanded upon the use of the kit. (Line 140-145)
- c) Immunocytochemistry and confocal microscopy:
Line 166: pls add abbreviation “MCTS” after multicellular tumor spheroids
- Fixed
- d) Trogocytosis Assay
Pls start the paragraph with a subject
- fixed
- Results:
- a) Pls number the figures always following the order of citation in the main text and also change the order of the panels in each figure based on what is cited first in the manuscript
- We have gone through all of our figures and addressed this problem.
- b) Pls better specify what is meant by Control CAR (paragraph 3.1)
- We have further expanded upon this in the text. (line 250-252)
- c) Fig.4: histogram graph in panel A is redundant to panel C as well as histogram graph in panel B to panel D. I suggest to display only graphs C and D.
- We thank the reviewer for the comment and have amended the figure.
- d) Fig. 7: these data should be supported by other experimental results. I suggest the authors to add cytotoxicity results of triple culture MCTS with not activated PBMCs to be performed and analyzed in comparison to PHA activated PBMC. Indeed the not activated PBMC could be a better resembling condition of in vivo status and data obtained from these experiments have to be compared to data obtained with PHA activated PBMC. Indeed, the higher cytotoxic activity, shown in panel B (i.e. CEA-CAR vs LS174T:BJ in fig 7B vs fig 6B), could be simply due a killing activity of effector cells present in activated PBMCs (add in panel D also the % of NK cells) or to secreted factors, cell contacts mediated by activated blood derived cells obtained from an in vitro setting difficult to reproduce in vivo.
- We have added an additional graph more clearly representing the controls made in Figure 7. We left out the PBMC control in the initial analysis because they were shown to be ineffective alone towards the MCTS. Additionally, when routine screening the PBMCS we also stained for CD56 in which the CD56 levels were below 10% in line with the phenotype data having roughly 90% CD3 positive cells. We have added the data with a comparison of PBMC alone vs NK92 with PBMCs showing that PBMC alone does not have any significant cytotoxic effect towards the triple culture model, however when combined with NK92 cells the cytotoxic effect is significantly enhanced confirming that the bulk cytotoxic effect seen can be coupled to the addition of CEA-NK92 cells.
Round 2
Reviewer 1 Report
Comments and Suggestions for Authors
Although the authors mentioned that NK-92 is widely used in clinical trials, and the authors' findings can be applied to human NK platform, still, human NK and NK-92 are completely different cells and responses to drugs are also quite different.
Please show that CEA-CAR, developed by the authors, is applicable to human NK cells. (eg. Can human NK cells be transduced appropriately by virus derived from CEA-CAR construct? Does CEA-CAR-human NK show reasonable tumor killing effect?)
Comments on the Quality of English Language.
Author Response
Reviewer 1:
Although the authors mentioned that NK-92 is widely used in clinical trials, and the authors' findings can be applied to human NK platform, still, human NK and NK-92 are completely different cells and responses to drugs are also quite different.
Please show that CEA-CAR, developed by the authors, is applicable to human NK cells. (eg. Can human NK cells be transduced appropriately by virus derived from CEA-CAR construct? Does CEA-CAR-human NK show reasonable tumor killing effect?)
- We acknowledge the reviewers comments and agree that there are many factors that differentiate primary human NK cells from the NK-92 cell line. However, NK-92 cells are an established NK cell model with many NK-cell characteristics that can be standardized. This particular aspect is one of the strengths of using an established model cell line to answer research questions and when making a standardized living drug compared to patient or donor derived primary NK cells, which come with donor variability that makes standardization harder. This aspect has been discussed and touched upon in the paper. In this paper, we wanted to focus on the NK-92 cell line as an alternative source to the primary NK cells because of the mentioned clinical advantages and therefore it is out of the scope of this paper to validate the CEA-Construct again using human NK cells.
Reviewer 2 Report
Comments and Suggestions for Authors
The revised version of the manuscript is suitable for publication.
Author Response
Reviewer 2:
The revised version of the manuscript is suitable for publication.
- We thank the reviewer for the comments and the scientific discourse. The suggested improvements and comments helped improve the final manuscript substantially.
Reviewer 4 Report
Comments and Suggestions for Authors
The authors have exhaustively answered almost all the requested questions
Author Response
Reviewer 4:
The authors have exhaustively answered almost all the requested questions
- We thank the reviewer for the comments and scientific discourse. The suggested improvements and comments helped improve the final manuscript substantially.